# Study of the Structure, Magnetic, Thermal and Electrical Characterisation of ZnCr_2_Se_4_: Ta Single Crystals Obtained by Chemical Vapour Transport

**DOI:** 10.3390/ma14112749

**Published:** 2021-05-22

**Authors:** Izabela Jendrzejewska, Tadeusz Groń, Piotr Kwapuliński, Joachim Kusz, Ewa Pietrasik, Tomasz Goryczka, Bogdan Sawicki, Andrzej Ślebarski, Marcin Fijałkowski, Josef Jampilek, Henryk Duda

**Affiliations:** 1Institute of Chemistry, University of Silesia in Katowice, 40-007 Katowice, Poland; ewa.pietrasik@us.edu.pl; 2Institute of Physics, University of Silesia in Katowice, 40-007 Katowice, Poland; tadeusz.gron@us.edu.pl (T.G.); joachim.kusz@us.edu.pl (J.K.); bogdan.sawicki@us.edu.pl (B.S.); andrzej.slebarski@us.edu.pl (A.Ś.); marcin.fijalkowski@us.edu.pl (M.F.); henryk.duda@us.edu.pl (H.D.); 3Institute of Materials Science, University of Silesia in Katowice, 40-007 Katowice, Poland; piotr.kwapulinski@us.edu.pl (P.K.); tomasz.goryczka@us.edu.pl (T.G.); 4Department of Analitycal Chemistry, Faculty of Natural Sciences, Comenius University, 842 15 Bratislava, Slovakia; josef.jampilek@gmail.com

**Keywords:** single crystals, semiconductors, antiferromagnetic, specific heat

## Abstract

The new series of single-crystalline chromium selenides, Ta-doped ZnCr_2_Se_4_, was synthesised by a chemical vapour transport method to determine the impact of a dopant on the structural and thermodynamic properties of the parent compound. We present comprehensive investigations of structural, electrical transport, magnetic, and specific heat properties. It was expected that a partial replacement of Cr ions by a more significant Ta one would lead to a change in direct magnetic interactions between Cr magnetic moments and result in a change in the magnetic ground state and electric transport properties of the ZnCr_2−x_Ta_x_Se_4_ (*x* = 0.05, 0.06, 0.07, 0.08, 0.1, 0.12) system. We found that all the elements of the cubic system had a cubic spinel structure; however, the doping gain linearly increased the ZnCr_2__−x_Ta_x_Se_4_ unit cell volume. Doping with tantalum did not significantly change the semiconductor and magnetic properties of ZnCr_2_Se_4_. For all studied samples (0 ≤ *x* ≤ 0.12), an antiferromagnetic order (AFM) below *T_N_*~22 K was observed. However, a small amount of Ta significantly reduced the second critical field (*H_c2_*) from 65 kOe for *x* = 0.0 (ZnCr_2_Se_4_ matrix) up to 42.2 kOe for *x* = 0.12, above which the spin helical system changed to ferromagnetic (FM). The *H_c_*_2_ reduction can lead to strong competition among AFM and FM interactions and spin frustration, as the specific heat under magnetic fields *H* < *H_c_*_2_ shows a strong field decrease in *T_N_*.

## 1. Introduction

Single crystals are required in many fields of modern technology and the electronic industry. They are used for lasers, as an optical component for spectroscopy, in oscillators, in light-emitting diodes and in innumerable other devices. Among seleno-spinel crystals such as AB_2_Se_4_ matrices, where A and B are the metallic ions occupying tetra- and octahedral sites, respectively, and matrices diluted with non-magnetic ions, due to the large cubic unit cell (about 10 Å), the effects of site disorder, lattice frustration and random distribution of spin interactions [1,2,3,4,5] create new potential applications.

The ZnCr_2_Se_4_ belongs to the group of seleno-spinels and crystallises in a cubic system (SG: Fd3m, No. 27), *a* = 10.4891 Å [6]. This compound is a *p*-type semiconductor with a helical magnetic structure. Magnetic interactions are the result of the existence of exchangeable Cr–Cr interactions competing with each other. The Cr–Se–Cr interaction causes the ferromagnetic (FM) ordering of the magnetic moments of chromium ions. In contrast, the Cr–Se–Se–Cr and Cr–Se–Zn–Se–Cr interactions lead to antiferromagnetic (AFM) ordering. The FM interactions are evidenced by the positive value of paramagnetic Curie–Weiss temperature *θ_CW_* = 115 K and AFM interactions are present below *T_N_* ≈ 21 K [7,8,9,10]. An increasing dc magnetic field shifts T_N_ to lower temperatures during a susceptibility peak in the paramagnetic region—to higher ones. Next, the first critical field *H_c_*_1_ values connected with a metamagnetic transition decrease slightly with temperature. The values of the second critical field, *H_c_*_2_, connected with the breakdown of the conical spin structure, drop rapidly with temperature, suggesting a spin frustration of the re-entrant type [2]. Low-angle neutron scattering (SANS) measurements showed the absence of any long-range magnetic order in the high-field (spin-nematic) phase, as well as the fact that all observed phase transitions were surprisingly isotropic concerning the field direction [11].

The introduction of the third cation to the ZnCr_2_Se_4_ crystal lattice, depending on many factors, may have a strong influence on changes in physicochemical properties. The most important factors are: (a) the size of the ion radius, (b) the position of the ion in the spinel structure (tetra- or octahedral), (c) the coordination number (CN), (d) the type of chemical bond [12]. Substitution of an additional cation into the ZnCr_2_Se_4_ matrix resulted in such phenomena as the appearance of spin glass [13,14,15], polaron conductivity [16], ferrimagnetism [17] and the enhancement of both FM [18,19] and AFM interactions [20,21].

This article describes the family of ZnCr_2_Se_4_ single crystals, doped with Ta, obtained by chemical vapour transport (CVT method), as well as their physicochemical properties. The purpose of the present study was to investigate the effect of the Ta ion admixture on the stability of the cubic symmetry and the physical (magnetic, electrical, thermal) properties of a ZnCr_2_Se_4_-based spinel. This work continues our previous works focusing on modifying ZnCr_2_Se_4_ properties by incorporating the additional *d*-electronic elements into the crystal lattice. In this study, we used a computer simulation of the mechanism of chemical vapour transport. To the best of our knowledge, this is the first demonstration of the synthesis of ZnCr_2_Se_4_ single crystals containing tantalum ions and their properties.

## 2. Materials and Methods

Commercially available high-purity elements Zn, Ta, Cr, Se (5N, Sigma Aldrich, Poznań, Poland) and anhydrous CrCl_3_ (Sigma Aldrich, Poznań, Poland) were used in the present study. Synthesis of ZnCr_2_Se_4_: Ta single crystals was carried out using a chemical transport reaction method. Chemical vapour transport is a method in which a solid substance is transferred using the reversible gaseous reaction from the *T*_1_ temperature (dissolution zone) to the *T*_2_ temperature (crystallisation zone). The partial equilibrium pressures of the components *p*_i_ depend on the value of the equilibrium constant *K*_a_ of the reaction, which can occur at a given temperature. The most significant changes in *p*_i_ occur when *K*_a_ reaches a value close to 1 (*logK*_a_ ≈ 0).

For this purpose, a thermodynamic model of ZnCr_2_Se_4_: Ta single-crystal growth was prepared. To determine the ability and conditions of transport reaction, a set of all the hypothetical reactions that may appear in the ZnSe-Ta-Se-CrCl_3_ system was created. These reactions, which were used to determine the dependence of *logK_a_* values on the temperature, were calculated using the HSC Chemistry computer programme (HSC Chemistry ver. 6.01, Release 2009, Metso Outotec Corporation, Helsinki, Finland) [22]. In this case, the modified chemical vapour transport method was used. The single crystals of ZnCr_2_Se_4_: Ta were grown from the binary selenide ZnSe, pure Ta and Se, and with CrCl_3_ as a transport carrier. Therefore, the hypothetical reaction set included the reactions of binary ZnSe, pure tantalum and selenium with CrCl_3_ and the dissociation products of CrCl_3_ (CrCl_4_ and Cl_2_). The ZnSe was obtained by the ceramic method [14,18]. Mixtures of the ZnSe, Ta, Se and CrCl_3_ were placed in quartz ampoules (length—200 mm, inner diameter—20 mm) evacuated to 10^−5^ mbarr. The ampoules were placed in a two-zone tubular furnace. The furnace was cooled for 24 h, after around 500 h of heating. The chemical composition of the obtained single crystals and their surfaces were studied by scanning electron microscopy (SEM) Jeol 6480 (JEOL USA, INC., Peabody, MA, USA), with a propertied energy-dispersive X-ray spectrometer (SEM/EDS). Four single crystals with differing tantalum content were chosen for X-ray diffraction (XRD) measurements (Table 1). The structural parameters were determined using a SuperNova X-ray diffractometer (Agilent, Oxfordshire, UK). The technical details and the computer programmes used are described in [21,23].

The electrical resistivity *ρ**(T)* was measured by the four-point DC method with an accuracy of around ±0.6% using a KEITHLEY 6517B Electrometer/High Resistance Meter (Keithley Instruments, LLC, Solon, OH, USA) in the temperature range of 77–400 K. A Quantum Design SQUID-based MPMSXL-5-type magnetometer (Quantum Design, San Diego, CA, USA) to determine the specific heat and magnetic parameters in the temperature range 10–300 K was used. Measurements were carried out at a magnetic field of 100 Oe for both in the ZFC (zero-field cooling) and FC (field cooling) mode. The Néel temperature and the critical fields were determined as the temperature corresponding to *dχ/dT* vs. *T* and d*M*/d*H* vs. *H*. The effective magnetic moment was determined using the following equation [24,25]:(1)$$\mu_{eff}=\sqrt{\frac{3k_{B}C}{N_{A}\mu_{B}^{2}}}≅2.828\sqrt{C}$$
where *k*_B_ is the Boltzmann constant, *N*_A_ is the Avogadro number, *μ_B_* is the Bohr magneton, and *C* is the molar Curie constant. The magnetic superexchange integrals for the first two coordination spheres *J*_1_ and *J*_2_ were calculated using the Holland and Brown equations [26].
(2)$$T_{N}=-5J_{1}+10J_{2},\;\theta =15J_{1}+90J_{2}$$

The methods used to determine the electrical properties and specific heat are described in detail in [17,18,20]. Thermogravimetry and differential scanning calorimetry (TG/DSC) were carried out using a Labsys Evo system, with a heating rate of 5 °C/min and in an inert gas atmosphere (Ar).

## 3. Results and Discussion

### 3.1. Growth of Single Crystals and Chemical Composition

Synthesis of the single crystals of ZnCr_2_Se_4_: Sn was carried out according to the reaction:4xZnSe + 4yHo + 4ySe + 2CrCl_3_→Zn_x_Ho_y_Cr_2_Se_4_ + 3xZnCl_2_ + 3yHoCl_2_, 
where: *y* = 0.1, 0.2, 0.3 and *x* = 1 − y.

We synthesised the samples with an amount of Ta ions higher than 0.3. For the samples with *y* = 0.4, 0.5, we did not observe single crystals in ampoules. One of the main reasons is that one end member of the Zn_x_Ta_y_Cr_2_Se_4_ series, i.e., TaCr_2_Se_4,_ does not exist. Based on the reactions presented in Figure 1, it can be assumed that volatile zinc and selenium compounds are formed in the ZnSe-Ta-Se-CrCl_3_ system, e.g., ZnCl_2_, Zn_2_Cl_4_, SeCl_2_, SeCl_4_ and Se_2_. Thermodynamic calculations showed that in the ZnSe-Ta-Se-CrCl_3_ system, ZnSe and Se are mainly transported by gaseous CrCl_3_ and CrCl_4_ (values of *logK_a_* are close to zero in the selected temperature range: 1000–1400 K; Figure 1 and Figure 2).

The transport reactions presented in Figure 2 show that volatile tantalum compounds can be formed in the ZnSe-Ta-Se-CrCl_3_ system, e.g., TaCl, TaCl_2_, TaCl_3_, TaCl_4_ and TaCl_5_. For CrCl_4_, *logK_a_* reaches values close to zero at lower temperatures than for CrCl_3_. It can therefore be assumed that the chemical transport for Ta occurs mainly with CrCl_4_. The *logK_a_* of the transport reactions with CrCl_3_ and CrCl_4_ for tantalum has values and shapes similar to the corresponding Zn and Se transport reactions (Figure 1). Thus, the conditions for the simultaneous transport of ZnSe, Ta and Se, and spinel crystallisation, are met. Calculated *logK_a_* values for transport reactions with chlorine are close to zero at higher temperatures. This indicates that chlorine is not involved in the transport process in the selected temperature range. Based on these calculations, the reaction conditions were chosen (temperatures of the dissolution and crystallisation zones and their difference). The crystallisation zone temperature was between 1070 and 1180 K. The melting zone temperature was in the range 1175–1233 K. Their difference was 50–70 K. Smaller differences in temperature (ΔT) are more favourable for the crystallisation process. The choice of reaction conditions was based on our experience with the growth of ZnCr_2_Se_4_ single crystals. The best results have been achieved using temperatures of the dissolution zone in the range 1123–1223 K. The choice of the reaction temperature is also limited by the thermal endurance of silica glass (~1500 K). These reaction conditions allowed us to obtain single crystals of good quality (Figure 3).

To determine the average chemical composition, measurements were carried out at 20 different locations of the single crystal. Each of the measuring areas was approximately 50 × 30 μm. Then, the average chemical composition was calculated. The error bar represents standard deviation. Relatively low values of standard deviation indicate good homogeneity of the chemical composition. Details of the reaction conditions and results of SEM measurements are presented in Table 1.

As shown in Table 1, the determined real chemical composition showed a lower amount of incorporated tantalum than the nominal composition. On the one hand, this may be due to the lower amount of tantalum transported to the crystallisation zone. On the other hand, the separation of tantalum from the system during the dissolution and crystallisation processes is likely.

### 3.2. Structural Study

The origin of the unit cell is agreed with the point $\bar{3}$*m* of the space group $Fd\bar{3}$*m* (No. 227). In a normal spinel, the Zn^2+^ ions occupy the tetrahedral position 8a: 1/8, 1/8, 1/8 (A site), and the Cr^3+^ ions occupy the position 16d: ½, ½, ½ (B site). To determine the cation distribution, two models were taken into account for each composition.

The first model of the structure refinement was considering the Ta ions’ presence in the A sites (tetrahedral) with coupled site occupancy factors (SOF) and constrained atomic displacements. Due to the strong correlation, the SOFs for Zn and Ta were refined separately in alternative calculations. The convergence caused the rejection of Ta ions from the tetrahedral sites.

The same procedure was applied to Ta ions occupied by the B sites (octahedral). This approach led to acceptable atomic displacement parameters and SOFs, which allowed for description of the general chemical formula for obtained single crystals as presented in Table 2 and Table 3. On the other hand, a slightly increased thermal shift at the B site may indicate a static disturbance in the position of the Cr^3+^/Ta^3+^ pseudo-ion, which is mainly caused by the difference in ionic charges and in ionic and covalent radii (r_i_(Zn^2+^) = 0.60 Å, r_i_(Cr^3+^) = 0.62 Å, r_i_(Ta^3+^) = 0.72 Å and R_c_(Zn^2+^) = 0.74 Å, R_c_(Cr^3+^) = 0.76 Å, R_c_(Ta^3+^) = 0.86 Å, respectively) [27]. Based on the structural study, the formula describing cation distribution in the system is: ZnCr_2-x_Ta_x_Se_4_, where x = 0.05, 0.06, 0.07, 0.08, 0.10 and 0.12, which is shown in Table 2 and Table 3. In these chemical compositions, the region of the solubility limit of tantalum in the ZnCr_2-x_Ta_x_Se_4_ system was found to be equal to 0.12. The structural study showed the obtained ZnCr_2-x_Ta_x_Se_4_ single crystals crystallise in a cubic system with space group $Fd\bar{3}$*m* (No.227, Z = 8). The lattice parameters of the obtained single crystals were larger than the lattice parameter of non-substituted ZnCr_2_Se_4_ (a = 10.489 Å) and increased with the rising amount of tantalum (Table 2).

Tantalum ions accommodated the octahedral sites, substituting Cr^3+^ ions. The enhancement of the lattice parameters was linear (Figure 4) and confirmed the presence of tantalum ions as Ta^3+^ in the crystal lattice of ZnCr_2_Se_4_. This phenomenon is consistent with the differences between the ionic radii of Zn^2+^, Ta^3+^ (in octahedral coordination) and Cr^3+^. The positional parameter of Se (*u*), which is a measure of the anion sublattice distortion from the cubic close packing, did not change significantly from the ideal value of *x* = 0.250. The tantalum amount did not influence the *u* parameter. The slight differences between the tested single crystals were in the range of the standard deviation (Table 3). The same was observed in the metal–metal and metal–selenium distances (Table 4), where the differences were insignificant. The structure refinement parameters are presented in Table 3, while the selected bond distances and angles are shown in Table 4.

### 3.3. Electrical and Magnetic Properties

The electrical measurements of ZnCr_2-x_Ta_x_Se_4_ (*x* = 0.5, 0.6, 0.7, 0.8, 0.10 and 0.12) showed the semiconducting properties and thermally activated conduction in the intrinsic region (200–400 K) with the activation energy of *E*_a_~0.22 eV. In the extrinsic region (77–125 K), the activation of electrical conductivity was not observed (Figure 5). The influence of anisotropy on the value of electrical resistivity in the studied spinel single crystals was not observed, as in the CuCr_1.6_V_0.4_Se_4_ spinel single crystal, in which the resistance was measured in the direction of [001] and [111] [28].

A similar dependence of electrical resistance *ρ**(T)* on temperature was observed by Watanabe [29] in ZnCr_2_Se_4_ single crystals measured in different crystallographic directions. No significant influence of the crystallographic orientation on the value of the specific electrical resistance was observed. This may have been because the spinel structure had a high cubic symmetry. The SANS measurements also confirmed the isotropic phase transitions with respect to the field direction in the monocrystalline ZnCr_2_Se_4_ spinel [11]. The observed *ρ**(T)* dependence in Figure 5 means that doping of the ZnCr_2_Se_4_ single crystals under study with tantalum did not affect the thermal activation of electrical conductivity.

The results of magnetic measurements of ZnCr_2-x_Ta_x_Se_4_ spinels are shown in Figure 6, Figure 7 and Figure 8 and Table 5. The dependence of the dc (in both ZFC and FC mode) showed an AFM order below the Néel temperature *T*_N_ ~22 K and FM short-range interactions, evidenced by the positive Curie–Weiss temperature *θ_CW_* < 100 K. The *T*_N_ temperature essentially does not depend on the concentration of Ta ion. On the other hand, these ions strongly influence the short-range magnetic interactions visible for different values of *θ_CW_* (Table 5).

The results mentioned above were confirmed by the negative values of the superexchange integral for the first coordination sphere (*J*_1_) and the positive values of the superexchange integral for the second coordination sphere (*J*_2_) (Table 5). The effective magnetic moment calculated from the magnetic susceptibility expansion at high temperature corresponded to the effective number of Bohr magnetons estimated for the corresponding content of *x* chromium ions, as the tantalum ion was in the +III oxidation state of the octahedral site in the spinel structure. However, an additional magnetic ion may appear when tantalum is in the +II or +IV oxidation state. Then, its magnetic susceptibility is in the order of 154 × 10^−6^ emu/mol at 293 K. No split of the ZFC–FC susceptibility was observed, which suggests a lack of spin frustration in the studied single crystals (Figure 6).

The magnetisation isotherms depicted in Figure 7 at 10, 20, 30, 40 and 60 K show characteristic behaviour for the AFM order of magnetic moments and a lack of saturation in the magnetic field of 70 kOe, below the Néel temperature. Magnetisation isotherms plotted at 10 K for all samples show two critical fields and an unexpected increase in the magnetic moment with the rise in the tantalum content (Figure 8). Arrows indicate the metamagnetic transition at the first critical field *H_c_*_1_, and the breakdown of the helical spin arrangement at the second critical field *H_c_*_2_ in Figure 8. The values of both critical fields weakly depend on the tantalum content in the sample. However, the tantalum ions strongly reduce the second critical field by approximately 20 kOe to that for the ZnCr_2_Se_4_ matrix [30], without significantly affecting the content of the first critical field. The result is a lack of magnetisation saturation above the second critical field due to weaker short-range ferromagnetic interactions.

### 3.4. Heat Capacity

Figure 9 and Figure 10 show the change in the temperature T_N_ with the increasing magnetic field B.

The specific heat experimental data obtained for the series of various Zn_1_Cr_2-x_Ta_x_Se_4_ (x = 0.05 − 0.12) compounds show very similar behaviour in the *C_B_(T)* characteristics obtained at different magnetic fields, which suggests that either the off-stoichiometry in Zn and/or Cr sites or Ta doping do not change the magnetic properties of the pristine ZnCr_2_Se_4_ compound. The magnetic field strongly decreases the temperature *T_N_* characteristic of the magnetic transitions and decreases the intensity of the *C* peak at T_N_. Figure 11 plots [T_N_(B) − T_N_(B = 0)]/T_N_(0) versus magnetic field *B*. It should be noted that [T_N_(B) − T_N_(B = 0)]/T_N_(0) ~B^2^, which is characteristic behaviour of the antiferromagnetic materials.

Figure 12 displays entropy S(T) = $\int_{0}^{T}\frac{C\left(T\right)}{T}dT$ calculated at the magnetic fields *B* = 0 and 5 T, respectively. For all investigated samples, the value of the magnetic and phonon contribution to the entropy at the ordering temperature *T_N_* and *B* = 0 is around 50% of the entropy expected, considering the magnetic contribution S_m_ = R × ln(2S + 1) = 11.52 J/Kmol_Cr_. The field dependence of *C* shown in Figure 12 is associated with the metamagnetic transition at the first critical field *H_c_*_1_. A very similar *C_B_(T)* behaviour was experimentally documented recently for a series of spinels, which did not show a change in the magnetic structure at *H_c_*_2_. Figure 12 also indicates magnetic contribution ΔS(T) = S(T, B = 0) − S(T, B = 5T) for temperatures *T* > *T_N_*, which seems to be characteristic of the family of doped ZnCr_2_Se_4_ and recently was discussed as an effect of spin fluctuations in the paramagnetic regime [14].

### 3.5. Thermal Analysis

For the thermal measurements, the single crystals with a more significant amount of Ta were chosen to show the changes in thermal properties under the influence of Ta. The thermogravimetric (TG) results and differential scanning calorimetry (DSC) measurements vs. temperature are shown in Figure 13 for the ZnCr_1.90_Ta_0.10_Se_4_ single crystals.

The TG curve shows a mass loss of 34%. This is a similar value to the pure ZnCr_2_Se_4_ (Table 6). The differential scanning calorimetry (DSC) curve showed a small endothermic peak at 710 °C for ZnCr_1.90_Ta_0.10_Se_4_, which confirmed that the melting effect occurs at a high temperature. These peaks indicate that, for the investigated crystals, the melting effect appears at high temperatures. However, a shift in the peak towards lower temperatures was observed. This was caused by additional cations (Ta^3+^), which caused stiffening of the crystal lattice. On the one hand, this effect could be caused by an increase in covalent binding. On the other hand, the presence of Ta^3+^ ions is a kind of “impurity”, which induces a change in the endothermic peak position. Similar behaviour was observed in the (Zn_1-x_Nd_x_) Cr_2_Se_4_ system [20].

## 4. Conclusions

In summary, we have presented new ZnCr_2_Se_4_ single crystals doped with tantalum. The single crystals ZnCr_2-x_Ta_x_Se_4_ (where x = 0.05, 06, 07, 08, 010 and 0.12) were obtained by chemical vapour transport (CVT method), examined and analysed with XRD, SEM, DSC/TG, SQUID and QD-PPMS. Structural, electrical, magnetic, specific heat and thermal measurements showed the spinel structure with the presence of tantalum ions as Ta^3+^ in the octahedral sites, semiconducting properties in the intrinsic region, AFM order below the Néel temperature of 22 K with the FM short-range interactions as well as a strong shift in the specific heat peak towards lower temperatures with an increase in the magnetic field and a weaker than expected magnetic and phonon contribution to entropy. Generally, the shift in the specific heat peak and the Néel temperature towards lower temperatures with increasing dc magnetic field is usually accompanied by a phase transition from a cubic to a tetragonal structure. In other words, a strong dc magnetic field broadens the temperature range of the paramagnetic state and the cubic structure of the spinel under study. The upper critical field indicates that the sample reached the state of FM order below the structural transition. The situation is similar in the case of the ZnCr_2_Se_4_ matrix. The greatest influence of tantalum was observed in the increase in magnetisation and reduction in the second critical field.

## Figures and Tables

**Figure 1 materials-14-02749-f001:**
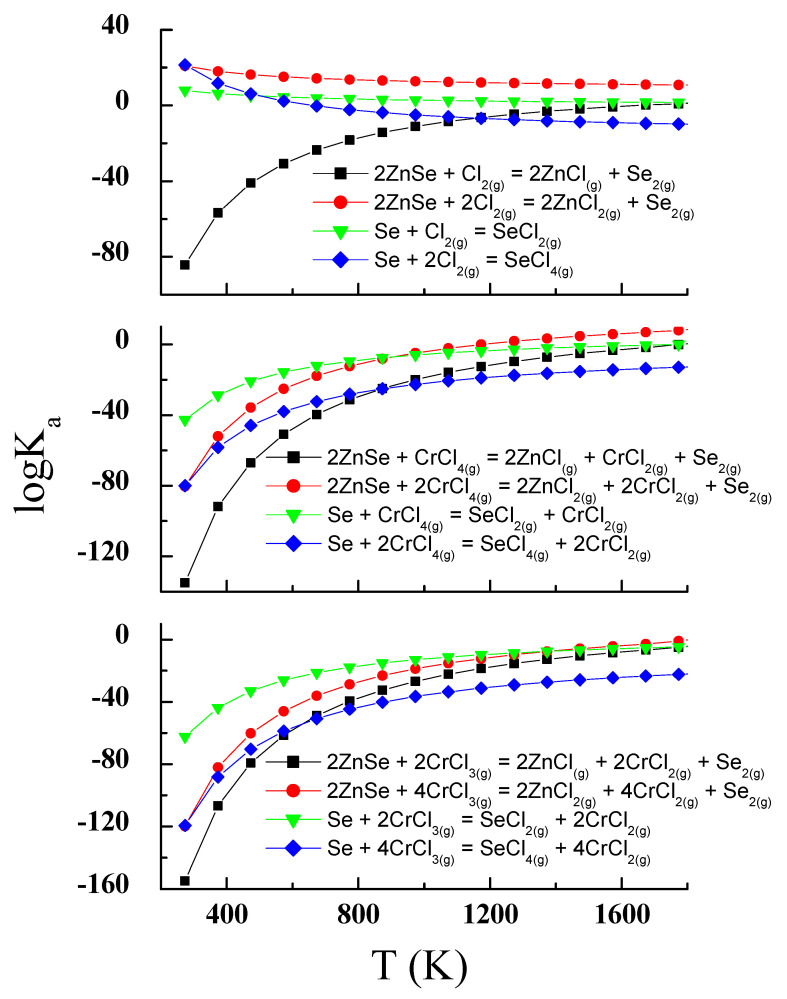
The dependence of the logK_a_ vs. temperature *T* for the ZnSe and pure Se transporting reactions.

**Figure 2 materials-14-02749-f002:**
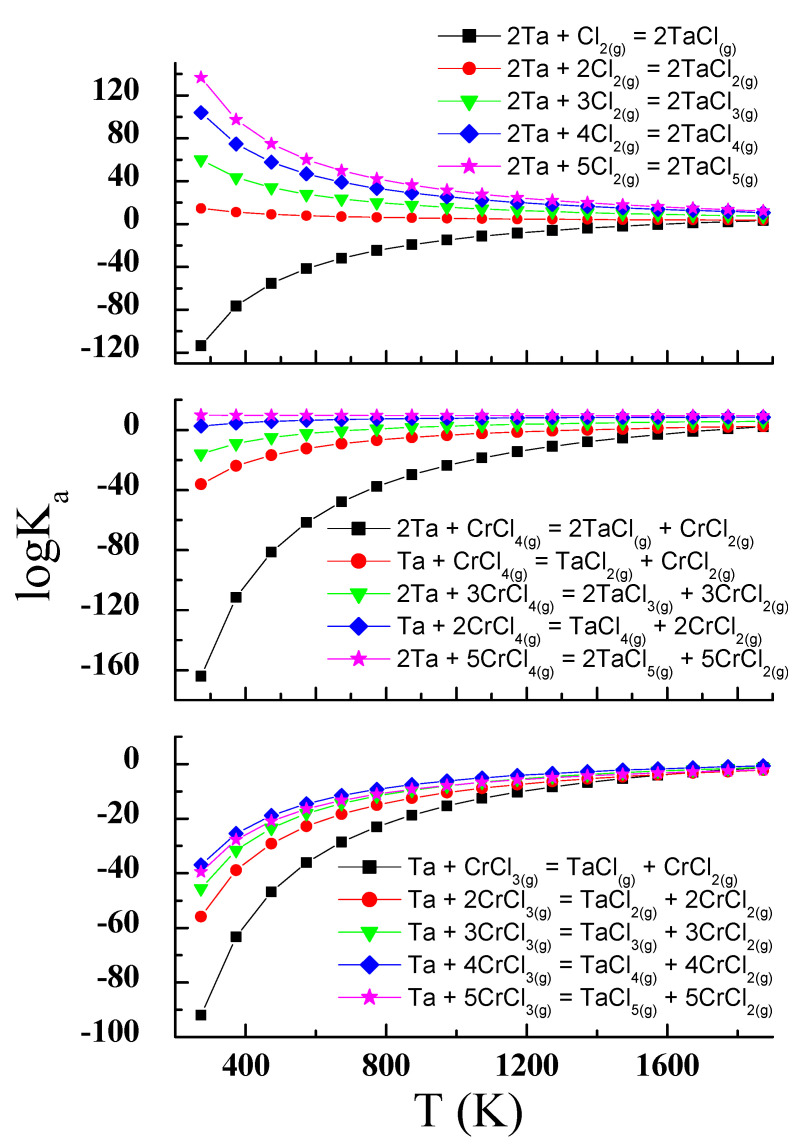
The dependence of the logK_a_ vs. temperature *T* for the pure Ta transporting reactions.

**Figure 3 materials-14-02749-f003:**
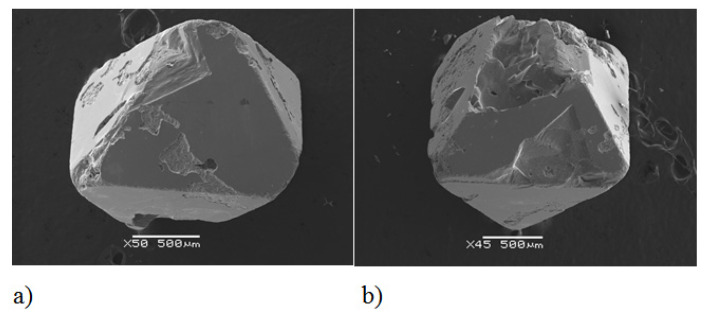
Examples of single crystals obtained in the ZnCr_2_Se_4_: Ta system: (**a**) Zn_1.03_Cr_1.96_Ta_0.05_Se_4_, (**b**) Zn_0.99_Cr_1.89_Ta_0.10_Se_4_.

**Figure 4 materials-14-02749-f004:**
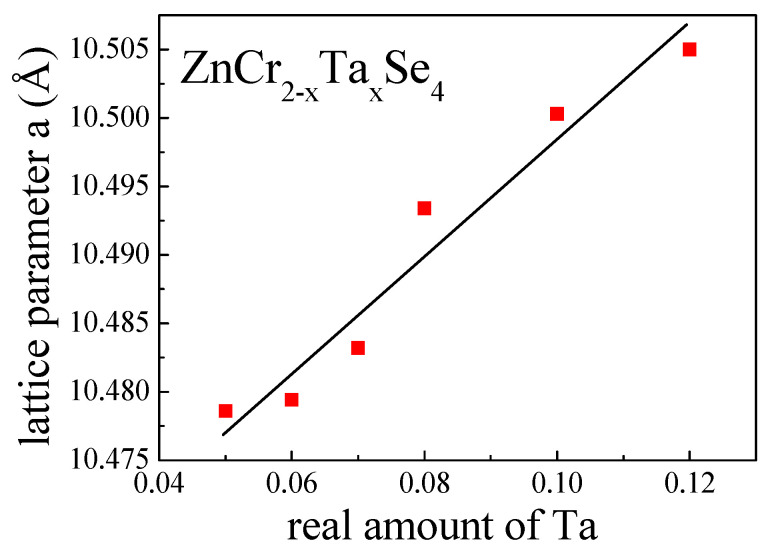
Dependence of the lattice parameters of ZnCr_2-x_Ta_x_Se_4_ single crystals on the amount of tantalum.

**Figure 5 materials-14-02749-f005:**
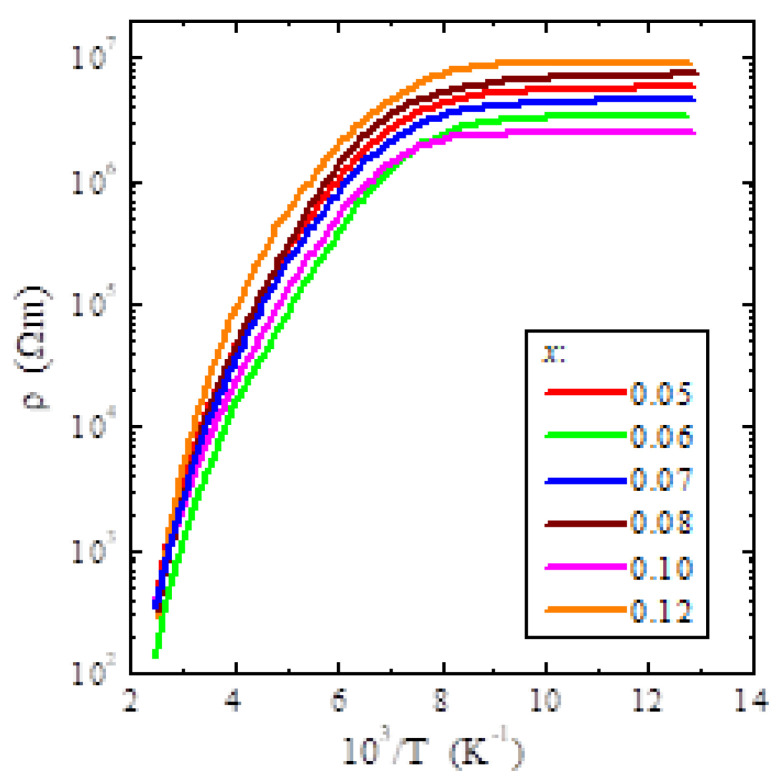
Electrical resistivity (ln*ρ*) vs. reciprocal temperature 10^3^/T of ZnCr_2-x_Ta_x_Se_4_ single crystals (where x = 0.05, 0.06, 0.07, 0.08, 0.10 and 0.12).

**Figure 6 materials-14-02749-f006:**
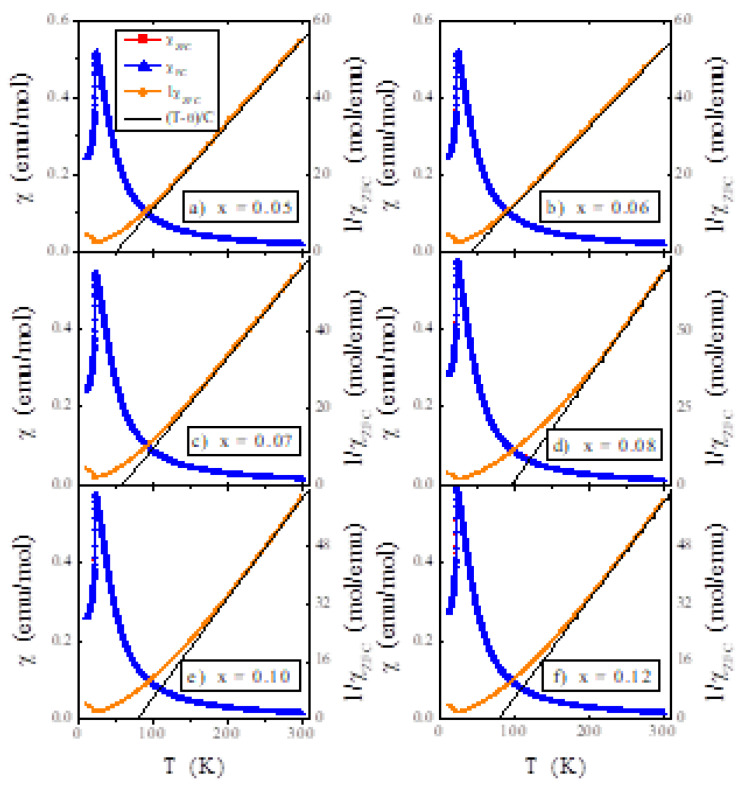
ZFC and FC magnetic susceptibility *χ* vs. temperature *T* of ZnCr_2−x_Ta_x_Se_4_ single crystals, where (**a**) 0.05, (**b**) 0.06, (**c**) 0.07, (**d**) 0,08, (**e**) 0.10 and (**f**) 0.12 recorded at *H* = 100 Oe. The solid (black) line, (T − θ)/C, indicates Curie–Weiss behaviour.

**Figure 7 materials-14-02749-f007:**
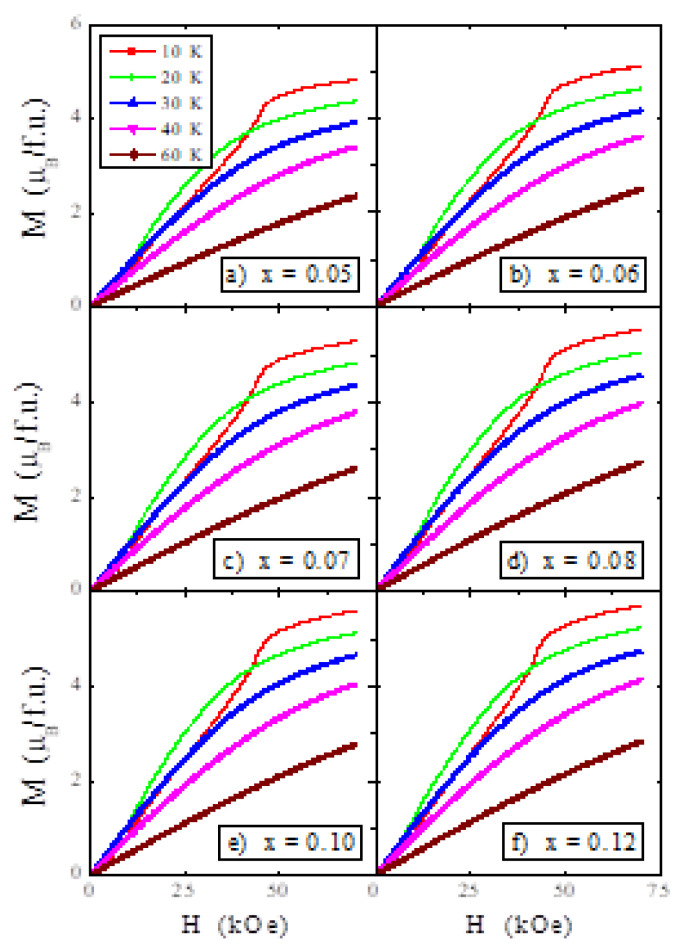
Magnetisation *M* vs. temperature *T* of ZnCr_2−x_Ta_x_Se_4_ single crystals, where (**a**) 0.05, (**b**) 0.06, (**c**) 0.07, (**d**) 0,08, (**e**) 0.10 and (**f**) 0.12 recorded at 10, 20, 30, 40 and 60 K.

**Figure 8 materials-14-02749-f008:**
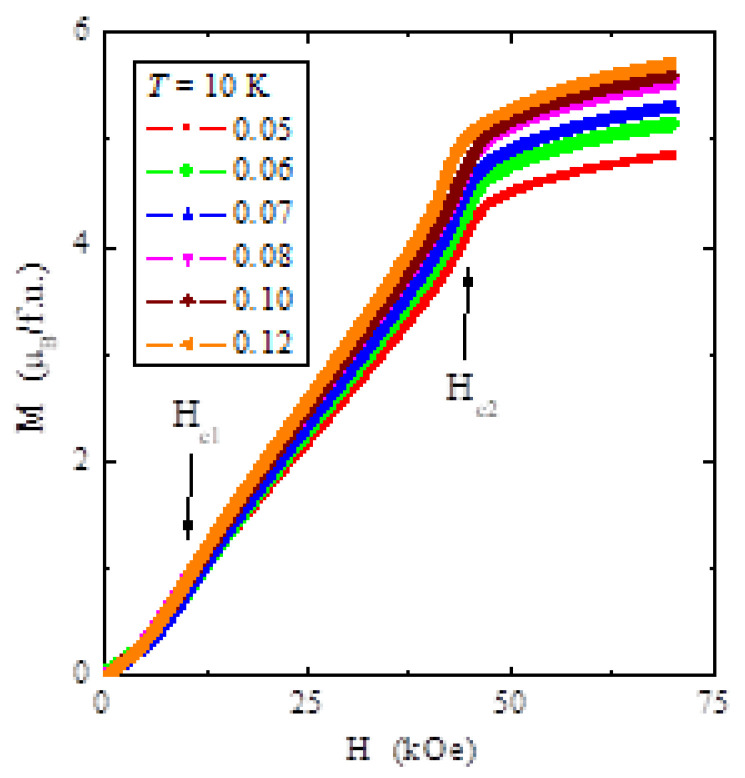
Magnetisation *M* vs. temperature *T* of ZnCr_2−x_Ta_x_Se_4_ single crystals (where x = 0.05, 0.06, 0.07, 0.08, 0.10 and 0.12) recorded at 10 K. Critical fields *H*_c1_ and *H*_c2_ are indicated by arrows.

**Figure 9 materials-14-02749-f009:**
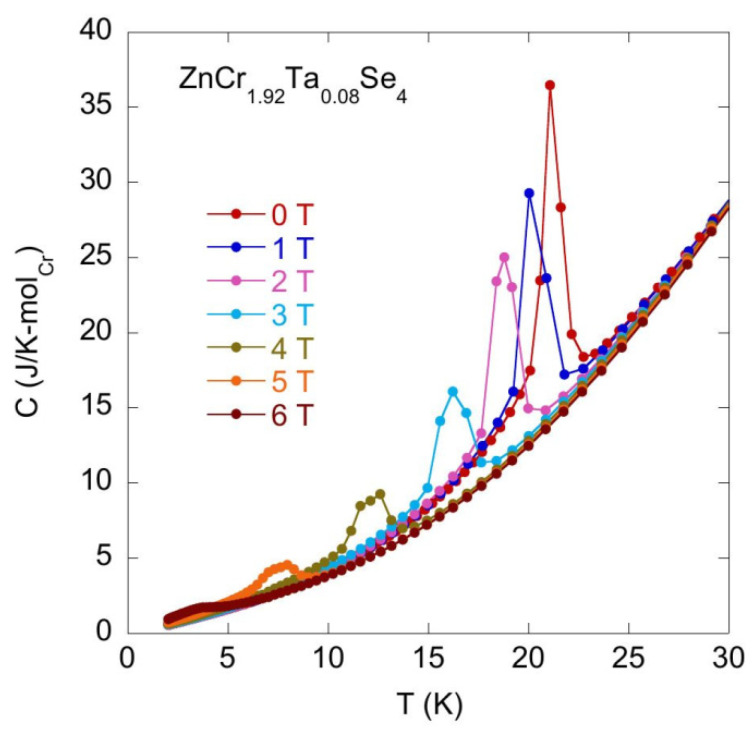
Specific heat *C* vs. temperature *T* measured for ZnCr_1.92_Ta_0.08_Se_4_ at zero and external magnetic fields.

**Figure 10 materials-14-02749-f010:**
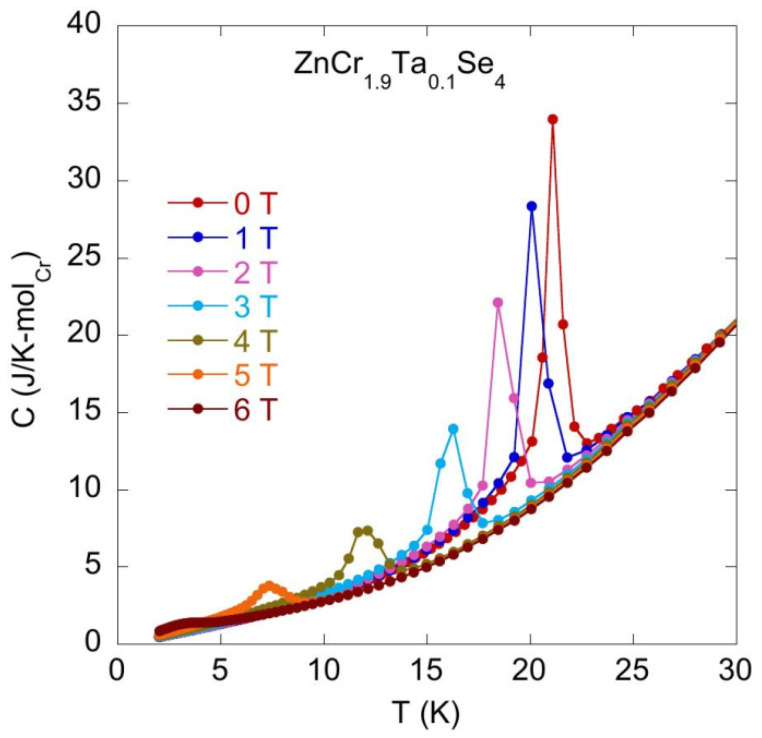
Specific heat *C* vs. temperature *T* measured for ZnCr_1.90_Ta_0.10_Se_4_ at zero and external magnetic fields.

**Figure 11 materials-14-02749-f011:**
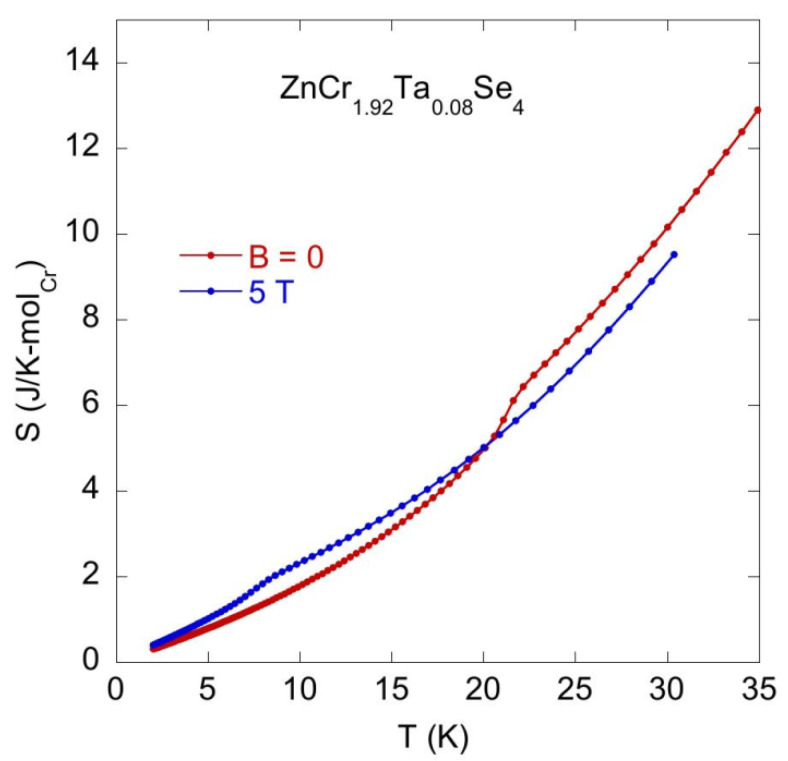
Entropy *S* per one Cr ion vs. temperature at the magnetic fields *B* = 0 and 5 T obtained for ZnCr_1.92_Ta_0.08_Se_4_.

**Figure 12 materials-14-02749-f012:**
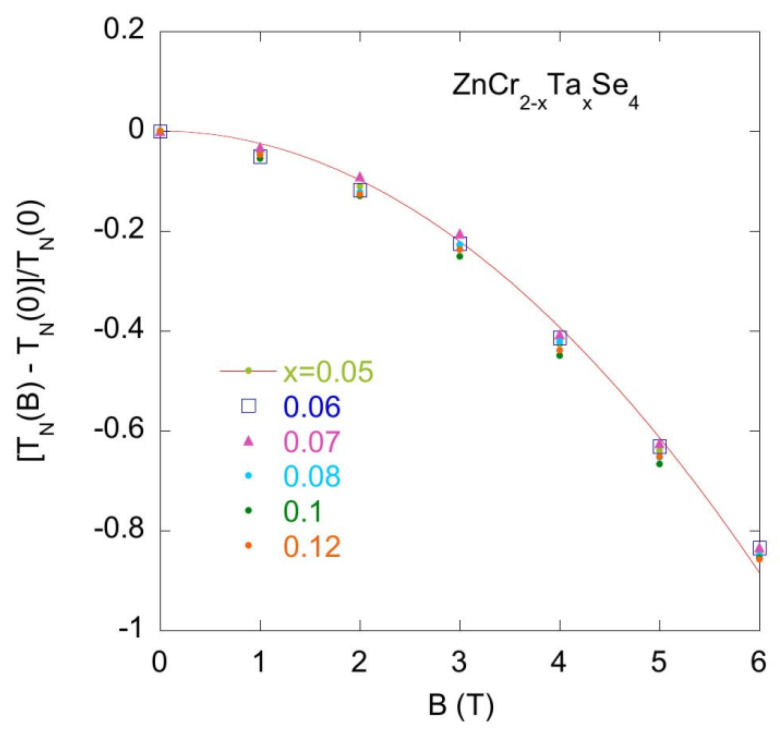
The expression [T_N_(B) − T_N_(B = 0)]/T_N_(0) vs. magnetic field for the series of ZnCr_2-x_Ta_x_Se_4_ samples (x = 0.05, 0.06, 0.07, 0.08, 0.1 and 0.12). The [T_N_(B) − T_N_(B = 0)]/T_N_(0) data are well approximated by the function f(B) = −aB^2^.

**Figure 13 materials-14-02749-f013:**
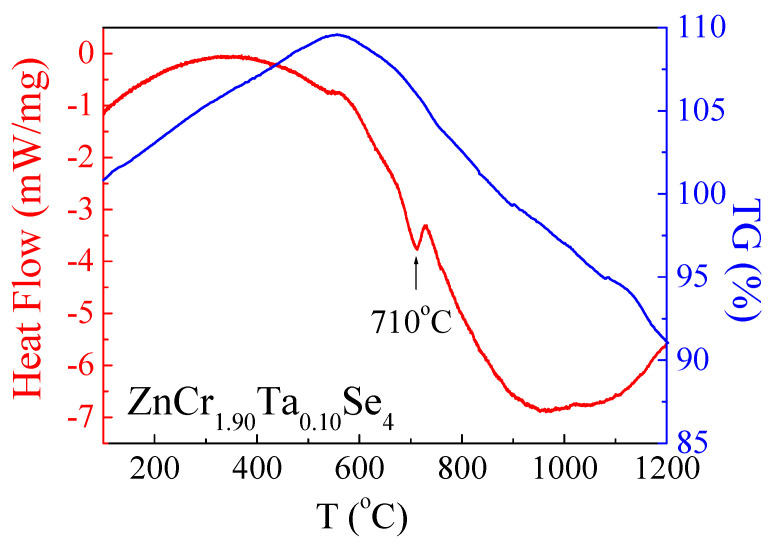
DSC/TG curves for ZnCr_1.90_Ta_0.10_Se_4_ single crystals.

**Table 1 materials-14-02749-t001:** Conditions of growth and chemical composition of ZnCr_2_Se_4_: Ta single crystals. *T*_d_ is the temperature of the dissolution zone, *T*_c_ is the temperature of the crystallisation zone and Δ*T* is the difference in temperature between the dissolution and the crystallisation zones.

No.	Nominal Formula	T_d_ (K)	T_c_ (K)	ΔT (K)	% Weight				Real Chemical Formula
					Zn	Cr	Ta	Se	
(1)	Zn_0.9_Ho_0.1_Cr_2_Se_4_	1223	1153	70	13.29 ± 0.09	20.66 ± 0.06	1.99 ± 0.01	64.06 ± 0.04	Zn_1.05_Cr_1.95_Ta_0.05_Se_4_
(2)	Zn_0.9_Ho_0.1_Cr_2_Se_4_	1133	1083	50	13.22 ± 0.08	20.53 ± 0.07	2.14 ± 0.02	64.11 ± 0.05	Zn_0.99_Cr_1.94_Ta_0.06_Se_4_
(3)	Zn_0.8_Ho_0.2_Cr_2_Se_4_	1203	1143	60	13.08 ± 0.06	20.37 ± 0.06	2.45 ± 0.05	64.10 ± 0.03	Zn_0.99_Cr_1.93_Ta_0.07_Se_4_
(4)	Zn_0.8_Ho_0.2_Cr_2_Se_4_	1203	1153	50	13.12 ± 0.02	20.12 ± 0.04	2.68 ± 0.06	64.18 ± 0.03	Zn_0.99_Cr_1.82_Ta_0.08_Se_4_
(5)	Zn_0.7_Ho_0.3_Cr_2_Se_4_	1203	1133	70	13.13 ± 0.03	19.75 ± 0.05	3.25 ± 0.03	63.87 ± 0.03	Zn_0.99_Cr_1.89_Ta_0.10_Se_4_
(6)	Zn_0.7_Ho_0.3_Cr_2_Se_4_	1203	1143	60	13.21 ± 0.02	19.53 ± 0.04	3.92 ± 0.04	63.34 ± 0.02	Zn_1.00_Cr_1.87_Ta_0.12_Se_4_

**Table 2 materials-14-02749-t002:** Structural parameters of the ZnCr_2-x_Ta_x_Se_4_ single crystals.

Chemical Formula	Lattice Parameter (Å)	Volume (Å^3^)	Density Calc. (Mg/m^3^)	Absorption Coeff. (mm^−1^)	Goodness of Fit on F^2^	R Parameters	
						R_1_	wR_2_
ZnCr_1.95_Ta_0.05_Se_4_	10.4786(12)	1150.56(4)	5.679	33.717	1.128	0.0122	0.0335
ZnCr_1.94_Ta_0.06_Se_4_	10.4794(3)	1150.82(2)	5.693	33.801	1.123	0.0145	0.0423
ZnCr_1.93_Ta_0.07_Se_4_	10.4832(9)	1152.08(9)	5.701	33.827	1.156	0.0195	0.0500
ZnCr_1.92_Ta_0.08_Se4	10.4938(12)	1155.56(2)	5.700	33.834	1.156	0.0179	0.0534
ZnCr_1.90_Ta_0.10_Se4	10.5003(15)	1157.73(3)	5.712	33.852	1.213	0.0155	0.0451
ZnCr_1.88_Ta_0.12_Se4	10.5069(18)	1159.90(3)	5.737	33.880	1.234	0.0164	0.0423

**Table 3 materials-14-02749-t003:** Atomic coordinates and equivalent isotropic displacement parameters of the ZnCr_2−x_Ta_x_Se_4_ (*x* = 0.05, 0.06, 0.07, 0.08, 0.10 and 0.12) single crystals. The Wyckoff positions of the atoms in the spinel structure are: Zn in 8 *b* (3/8, 3/8, 3/8); Cr/Ta in 16*c* (½, ½, ½) and Se in 32 *e* (x, x, x).

Spinel	Anion Parameter *u*	SOF in A Site	SOF in B Site	U_iso_ (Å × 10^3^)		
		Zn	Cr/Ta	Zn	Cr/Ta	Se
ZnCr_1.95_Ta_0.05_Se_4_	0.2594(1)	1.0	1.951:0.049(9)	8.73(15)	5.77(15)	6.13(12)
ZnCr_1.96_Ta_0.04_Se_4_	0.2594(1)	1.0	1.961:0.039(9)	8.83(15)	5.65(15)	6.24(12)
ZnCr_1.93_Ta_0.07_Se_4_	0.2594(1)	1.0	1.929:0.071(9)	8.83(15)	5.65(15)	6.24(12)
ZnCr_1.92_Ta_0.08_Se_4_	0.2594(2)	1.0	1.919:0.081(1)	8.24(18)	7.14(17)	7.57(12)
ZnCr_1.90_Ta_0.10_Se_4_	0.2594(2)	1.0	1.902:0.098(2)	8.80(17)	6.15(16)	6.27(12)
ZnCr_1.88_Ta_0.12_Se_4_	0.2594(1)	1.0	1.879:0.121(3)	8.67(12)	5.98(14)	6.54(12)

**Table 4 materials-14-02749-t004:** Selected interatomic distances (Å) and bond angles (deg) of the ZnCr_2-x_Ta_x_Se_4_ spinel single crystals.

Spinel	Bond Distances		Bond Angles	
	Zn-Se	Cr/Ta-Se	Se-Zn-Se	Se-Cr/Ta-Se
ZnCr_1.95_Ta_0.05_Se_4_	2.4388(3)	2.5252(2)	109.5(0) × 6	180.0(0) × 3
				94.543(7) × 6
				85.457(9) × 6
ZnCr_1.94_Ta_0.06_Se_4_	2.4401(3)	2.5248(2)	109.5(0) × 6	180.0(0) × 3
				94.574(7) × 6
				85.4426(9) × 6
ZnCr_1.93_Ta_0.07_Se_4_	2.4407(4)	2.5253(2)	109.5(0) × 6	180.0(0) × 3
				94.575(9) × 6
				85.425(9) × 6
ZnCr_1.92_Ta_0.08_Se_4_	2.4450(4)	2.5299(2)	109.5(0) × 6	180.0(0) × 3
				94.575(8) × 6
				82.425(8) × 6
ZnCr_1.90_Ta_0.10_Se_4_	2.4454(4)	2.55137(2)	109.5(0) × 6	180.0(0) × 3
				94.555(8) × 6
				85.445(8) × 6
ZnCr_1.88_Ta_0.12_Se_4_	2.4482(4)	2.55543(2)	109.5(0) × 6	180.0(0) × 3
				94.575(8) × 6
				85.425(8) × 6

**Table 5 materials-14-02749-t005:** Magnetic parameters of ZnCr_2-x_Ta_x_Se_4_ spinels—*C* is the Curie constant, *T_N_* is the Néel temperature, *θ_CW_* is the Curie–Weiss temperature, *µ_eff_* is the effective magnetic moment, *p*_eff_ is the effective number of Bohr magnetons, *J*_1_ and *J*_2_ are the superexchange integrals for the first two coordination spheres, *H*_c1_ and *H*_c2_ are the critical fields. Experimental data for ZnCr_2_Se_4_ were taken from Ref. [30] for comparison.

Spinel	C (emu × K/mol)	T_N_ (K)	θ_CW_ (K)	µ_eff_ (µ_B_/f.u.)	M_(10K)_ (µ_B_/f.u.)	p_eff_	J_1_ (K)	J_2_ (K)	H_c1_ (kOe)	H_c2_ (kOe)
ZnCr_2_Se_4_	4.08	21	90	5.71	6.0	5.477	−1.65	1.28	10.0	65.0
ZnCr_1.95_Ta_0.05_Se_4_	4.563	22.7	50.3	6.041	4.85	5.408	−2.57	0.99	10.0	43.5
ZnCr_1.94_Ta_0.06_Se_4_	4.944	22.6	41.5	6.288	5.13	5.394	−2.70	0.91	10.4	43.9
ZnCr_1.93_Ta_0.07_Se_4_	4.278	22.4	58.8	5.849	5.30	5.381	−2.38	1.05	10.7	42.7
ZnCr_1.92_Ta_0.08_Se_4_	2.971	22.3	96.1	4.875	5.54	5.367	−1.74	1.36	10.7	45.6
ZnCr_1.90_Ta_0.10_Se_4_	3.630	22.5	79.1	5.388	5.61	5.339	−2.06	1.22	10.2	43.3
ZnCr_1.88_Ta_0.12_Se_4_	3.696	21.9	78.4	5.437	5.71	5.310	−1.98	1.20	10.4	42.2

**Table 6 materials-14-02749-t006:** Parameters determined from TG and DSC analysis for ZnCr_1.90_Ta_0.10_Se_4_ single crystals. Data for pure ZnCr_2_Se_4_, presented in [20], are shown for comparison.

Ta Content	Weight Loss (%)	Onset (^o^)	Endset (^o^)	Peak Minimum (^o^)	Peak Height (mW)	Peak Area (J)	Enthalpy (J/g)
0.0	35	735	771	755	6.47	1.09	51.9
0.10	34	693	725	710	4.42	1.83	117.3

## Data Availability

The data presented in this study are available on request from the corresponding author.

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
