# Peer review of "Study of the Structure, Magnetic, Thermal and Electrical Characterisation of ZnCr2Se4: Ta Single Crystals Obtained by Chemical Vapour Transport"

_materials, 2021, doi:10.3390/ma14112749_

Round 1
Reviewer 1 Report
This is a very detailed article on the synthesis and properties of Zn Cr2Se4 crystals doped with Ta. There is a huge amount of work that has gone into what is presented in this manuscript. The key details that are required however need to be provided in a more digestible manner for the readers.
I recommend the publication of this manuscript if all of the following can be addressed in the revised manuscript.
- The Abstract is full of acronyms/abbreviations. It is a general rule that no acronym/abbreviations are used in an abstract, especially if they are not defined prior to use. These cannot be assumed to be known by readers. Also, the sentences need to be broken up here to avoid lengthy sentences.
- The motivation for the work is not brought out very clearly or convincingly and this needs improving in the introduction section. Why is Ta chosen specifically as a substituent?
- A reference needs to be provided for the “HSC Chemistry computer programme”
- The results of Fig.1 are not adequately discussed-it is difficult to note whether any crucial decisions regarding the temperature choices. Were made based on these plots. Such a discussion would be relevant here.
- More details regarding the reaction conditions are needed. As this is a paper which presents results on a variety of compositions of crystals, it will be useful for the readers to learn what was attempted and what was obtained at the different conditions used. The authors should at least provide the exact conditions which are conducive for the growth of the best crystals. Currently the temperature gradient used for the CVT is just given as a ‘range’. More details of the temperatures of the hot and cold ends attempted and what gave the best results are required.
- Are the chemical compositions listed in Table 1, the nominal starting compositions or the estimated compositions of the resulting crystals. I do not see any details of what the correlation between the two is, i.e., how did the Ta content relate to the initial starting compositions and how did this vary in the crystals obtained?
- Similarly in Table 2 and the results plotted in Fig.4, is the composition the estimated/resultant compositions of the Ta in the crystals? It is also not clear what the “XRD composition” listed in Table 3 is. Is this estimated from the XRD occupancy refinements? All of these different ‘compositions’ have to be explained or the same nomenclature used for all the data.
- Table 3 has an error: column showing Cr/Ho -should it read Cr/Ta?
- There is a lot of data presented on both the magnetic susceptibility and the magnetization isotherms for a number of crystals of varying compositions. The behaviour in most of them is similar and unless there are marked changes, there is no point showing so many sets of data in a paper (this is not a student’s Thesis). They are quite repetitive and can be cut out, showing just one typical plot- and retain plots such as Fig.8.
- For the data shown in Fig 13, why was this particular crystal chosen for the DSC/TGA study?
- The conclusions again run into lengthy sentences, please break this up for the ease of the reader.
The manuscript will be much improved once these above changes have been incorporated.
Author Response
We thank the Reviewers’ for their careful reading of the paper and for their constructive remarks. In order to take into account the latter, the paper has been revised. All changes are marked in yellow.
Reviewer #1:
- The Abstract is full of acronyms/abbreviations. It is a general rule that no acronym/abbreviations are used in an abstract, especially if they are not defined prior to use. These cannot be assumed to be known by readers. Also, the sentences need to be broken up here to avoid lengthy sentences.
A: The acronyms have been removed.
- The motivation for the work is not brought out very clearly or convincingly and this needs improving in the introduction section. Why is Ta chosen specifically as a substituent?
A: The purpose of the present study was to investigate Ta ions admixture on the stability of the cubic symmetry and the physical (magnetic, electrical, thermal) properties of ZnCr2Se4 – based spinel. It is a continuation of the work series concerning the mechanism of incorporation of transition metals into the chromium selenide spinels, and the role of this admixture in modifying the physical and transport properties. Spinel compounds doped with Ta are unknown.
- A reference needs to be provided for the “HSC Chemistry computer programme”
A: The information has been added to the text.
- The results of Fig.1 are not adequately discussed-it is difficult to note whether any crucial decisions regarding the temperature choices. Were made based on these plots. Such a discussion would be relevant here.
A: The information has been added to the text.
- More details regarding the reaction conditions are needed. As this is a paper which presents results on a variety of compositions of crystals, it will be useful for the readers to learn what was attempted and what was obtained at the different conditions used. The authors should at least provide the exact conditions which are conducive for the growth of the best crystals. Currently the temperature gradient used for the CVT is just given as a ‘range’. More details of the temperatures of the hot and cold ends attempted and what gave the best results are required.
A: The details of reactions conditions have been added to Table 1. For some transition metal ions, the values of site-preference- energy (Ev) to occupy octahedral are known (Cr3+: 69.5kJmol, Ni2+: 37.7kJ / mol). Based on these values, it can be assumed where the ion will be incorporated. The Cr3+ ions in selenium spinels always occupy octahedral positions, as they have the highest Ev values. And for example, Zn2+ ions have one of the smallest Ev values: -132.3kJ mol. And they always occupy tetrahedral positions. There are known deviations from this rule because, for example, In3+ ions with Ev = -168.3kJ/mol, theoretically should occupy tetrahedral positions and are very eager to locate in both tetrahedral and octahedral positions [1]. In the case of the ZnCr2Se4:Ta system, it was not known in what positions the tantalum ions would be incorporated, and the weight-outs were prepared according to the standard CVT reaction. A similar phenomenon was observed for the systems: ZnCr2Se4:Sn [2], ZnCr2Se4:Dy [3], ZnCr2Se4: Ho [4], ZnCr2Se4:Ni [5], CuCr2Se4:Ni [6].
- Maciążek, T. GroÅ„, H. Duda, A. WaÅ›kowska, I. Jendrzejewska, E. Malicka, E. Augustyn, S. Mazur, ” Influence of the cation substitution on the electrical conductivity of the n-type CuxInyCrzSe4 spinels”, Journal of Alloys and Compounds, 480 (2009) 60-62.
2. I. Jendrzejewska, T. Groń, J. Kusz, M. Żelechower, E. Maciążek, A. Ślebarski, M. Fijałkowski, "Spin-glass-like behaviour in tin doped ZnCr2Se4 single crystals", Journal of Alloys and Compounds, 635 (2015) 238-244.
- I. Jendrzejewska, T. GroÅ„, E. Maciążek, H. Duda, M. Kubisztal, A. Åšlebarski, E. Pietrasik, M. FijaÅ‚kowski, “Specific heat and magnetic properties of single-crystalline ZnxDyyCrzSe4 spinels”, Journal of Magnetism and Magnetic Materials (2016). 407, 122-128.
4. I. Jendrzejewska , T. GroÅ„, J. Kusz, J. Goraus, Z. Barsova, E. Pietrasik, J. Czerniewski, T. Goryczka, M. Kubisztal, „Growth, structure and physico-chemical properties of monocrystalline ZnCr2Se4:Ho prepared by chemical vapour transport”, Journal of Solid State Chemistry 281 (2020) 121024
- I. Jendrzejewska, A. Waśkowska, T. Mydlarz, " Influence of nickel substitution on the cation distribution and magnetic properties of ZnCr2Se4", Journal of Alloys and Compounds 327 (2001) 73-77.
- I. Jendrzejewska, P. Zajdel, T. GroÅ„, H. Duda, T. Mydlarz, “Effect of Ni doping on magnetic and electrical properties of CuCr2Se4 single crystals”, Journal of Alloys and Compounds, 2014, 593, 158-162
- Are the chemical compositions listed in Table 1, the nominal starting compositions or the estimated compositions of the resulting crystals. I do not see any details of what the correlation between the two is, i.e., how did the Ta content relate to the initial starting compositions and how did this vary in the crystals obtained?
A: The real amount of tantalum is lower than nominal. It is probably caused by a lower amount of transported Ta ions from the dissolution zone to the crystallization zone. On the other hand, the process of “escaping” Ta ions from the reaction system and forming other compounds is likely. For bigger amounts of Ta (x = 0.4, 0.5) growth of crystal growth was not observed. The region of the solubility limit of tantalum in the ZnCr2Se4:Ta system was found to be equal to 0.12. Similar phenomena were observed, for instance, in the systems mentioned above.
- Similarly in Table 2 and the results plotted in Fig.4, is the composition the estimated/resultant compositions of the Ta in the crystals? It is also not clear what the “XRD composition” listed in Table 3 is. Is this estimated from the XRD occupancy refinements? All of these different ‘compositions’ have to be explained or the same nomenclature used for all the data.
A: The chemical compositions obtained from SEM measurements are in good agreement with the structural data. Tables 2 - 5 show the refined chemical compositions, taking into account the occupations in tetrahedral and octahedral positions. Table 3 has been corrected. The parentheses have been removed. However, it is worth noticing that the formula (A)[B2]X4 indicates that A atoms occupy tetrahedral sites, and B atoms occupy octahedral sites.
- Table 3 has an error: column showing Cr/Ho -should it read Cr/Ta?
A: It has been corrected.
- There is a lot of data presented on both the magnetic susceptibility and the magnetization isotherms for a number of crystals of varying compositions. The behaviour in most of them is similar and unless there are marked changes, there is no point showing so many sets of data in a paper (this is not a student’s Thesis). They are quite repetitive and can be cut out, showing just one typical plot- and retain plots such as Fig.8.
A: The panels in Figs. 6 and 7 have been arranged in a more compact manner. The arrangement of the panels in Fig. 7 with an increase in the tantalum content shows the reader an increasing value of magnetization and a lack of saturation.
- For the data shown in Fig 13, why was this particular crystal chosen for the DSC/TGA study?
A: This explanation has been added to the text. We usually choose one crystal because the thermal analysis destroys the sample.
- The conclusions again run into lengthy sentences, please break this up for the ease of the reader.
A: The conclusions were clarified. The following text was added: “Generally, the shift of the specific heat peak and the Neel temperature [2-5,30] towards lower temperatures with increasing dc magnetic field is usually accompanied by a phase transition from cubic to the tetragonal structure. In other words, a strong dc magnetic field broadens the temperature range of the paramagnetic state and the cubic structure of the spinel under study. The upper critical field informs that the sample has reached the state of FM order below the structural transition. The situation is similar in the case of the ZnCr2Se4 matrix.”
Reviewer 2 Report
Generally, the manuscript is well written, however there are spelling mistakes at some places in the manuscripts. I consider that the content of this article is relevant to Materials journal and has considerable high scientific soundness.
Abstract must be enriched via valuable results which pave the way for understanding the audiences. Also, please include your recommendations and future prospects. Please define the used abbreviations.
The introduction section is very short and poorly described. It doesn't totally present the reference to the manuscript scope. The significance of this study should be more emphasize in the introduction.
Results and discussions. The quality and interpretation of obtained results should be improved. Formatting and quality of the figures should be considerably improved (the plots are not of the same size, some of them are too large.
Please be sure that the Conclusions section not only recapitulate the key findings of your work, but also explain the specific ways in which this work fundamentally advances the field relative to prior literature.
References. There is a tendency of SELF-citation. This should be reconsidered.
English of the paper is rather good – in my opinion the language of the paper should be a little improved. I am asking for corrections by a native speaker.
I consider that the article can be accepted for publication only after a major revision.
Author Response
We thank the Reviewers’ for their careful reading of the paper and for their constructive remarks. In order to take into account the latter, the paper has been revised. All changes are marked in yellow.
Reviewer #2:
- Generally, the manuscript is well written, however there are spelling mistakes at some places in the manuscripts. I consider that the content of this article is relevant to Materials journal and has considerable high scientific soundness.
A: Spelling errors have been corrected.
- Abstract must be enriched via valuable results which pave the way for understanding the audiences. Also, please include your recommendations and future prospects. Please define the used abbreviations.
A: The abstract has been corrected, the abbreviations used have been defined.
- The introduction section is very short and poorly described. It doesn't totally present the reference to the manuscript scope. The significance of this study should be more emphasize in the introduction.
A: The introduction has been extended as to the scope of reference and the meaning of this study. The following text was added: “Among seleno-spinel crystals such as AB2Se4 matrices, where A and B are the metallic ions occupying tetra- and octahedral sites, respectively, and matrices diluted with non-magnetic ions, due to the large cubic unit cell (about 10 Å), the effects of site disorder, lattice frustration and random distribution of spin interactions [1–5] create new potential applications.”; …, “An increasing dc magnetic field shifts TN to lower temperatures during a susceptibility peak in the paramagnetic region—to higher ones. Next, the first critical field Hc1 values connected with a metamagnetic transition decrease slightly with temperature, while the values of the second critical field Hc2, connected with the breakdown of the conical spin structure, drop rapidly with temperature suggesting a spin frustration of the re-entrant type [2[. Low-angle neutron scattering (SANS) measurements showed the absence of any long-range magnetic order in the high-field (spin-nematic) phase, as well as the fact that all observed phase transitions were surprisingly isotropic concerning the field direction [11].”
- Results and discussions. The quality and interpretation of obtained results should be improved. Formatting and quality of the figures should be considerably improved (the plots are not of the same size, some of them are too large.
A: The quality and interpretation of the results obtained have been improved. The formatting of the drawings, quality and size were done.
- Please be sure that the Conclusions section not only recapitulate the key findings of your work, but also explain the specific ways in which this work fundamentally advances the field relative to prior literature.
A: The conclusions were clarified. The following text was added: “Generally, the shift of the specific heat peak and the Neel temperature [2-5,30] towards lower temperatures with increasing dc magnetic field is usually accompanied by a phase transition from cubic to a tetragonal structure. In other words, a strong dc magnetic field broadens the temperature range of the paramagnetic state and the cubic structure of the spinel under study. The upper critical field informs that the sample has reached the state of FM order below the structural transition. The situation is similar in the case of the ZnCr2Se4 matrix.”
- References. There is a tendency of SELF-citation. This should be reconsidered.
A: The bibliography has been expanded. As the literature on the study of matrix-based compounds is abundant, it has necessarily been limited to the most relevant items.
- English of the paper is rather good – in my opinion the language of the paper should be a little improved. I am asking for corrections by a native speaker.
A: The article has been reviewed and corrected by an English teacher.
Reviewer 3 Report
In this article, Izabela Jendrzejewska et al. have synthesized ZnCr2-xTaxSe4 samples with different Ta doping levels using chemical vapor transport method. A thermodynamic growth model is proposed and the synthesis details are presented. Subsequently SEM/EDX, XRD, resistivity, SQUID, DSC/TG, and heat capacity measurement are performed/analyzed on these series of samples. Their main findings are 1) Series of tantalum doped samples are obtained, 2) tantalum doping reduces the upper critical field compared to parent compound but does not change TN, it also increases magnetization in the paramagnetic state 3) heat capacity shows phase transition shifting to lower temperature with increase field and smaller magnetic/phonon contribution to the entropy.
Overall this article is fluently written (outside of a few grammatical errors) and the results are clearly presented. I would recommend this article to be published if some experiment details and analysis are more clearly spelled as the comments shown below:
- In the introduction part (line 35), the application of ZnCr2Te4 as a magnetoelectric compound merits more reference and introduction before discussing the underlying physics, so the article can be more accessible to non-specialist readers.
- In table 1, authors used SEM/EDX to establish the chemical composition of the samples. For sample #2 to #6, the Zn composition are within 1% of the nominal doping and Ta composition are established accordingly, however for sample #1 the Zn is 5% more than nominal doping, which indicates the error bar of SEM is up to 5%, and it will render the Ta doping level up to 6% (0.12) in question. I do understand the XRD data shows systematic change of lattice parameters, I recommend authors either clarify the limit of SEM in a more detailed manner or even better, present ICP results.
- In part 3.4, resistivity and SQUID results are presented, however the setup of the measurement, especially the sample orientation information is missing. In parent compound, previous neutron scattering results indicate an incommensurate helical structure with different domains, under small magnetic field, domain selection occurs, so I naively figure this sample is not isotropic. Since the article is about single crystals, orientation of the measurement needs clarification.
- In figure 8 and table 5, how are the HC1 and HC2 established? By taking first derivative or fitting etc.?
- In part 3.5 where heat capacity measurement is presented, one main conclusion is that a phase transition shifting to lower temperature with increase field. Is this phase transition associated with the upper critical field? How does the result compare with undoped parent compound?
- In line 246, could authors elaborate more on “much smaller than expected” a bit more? What is the underlying physics/reference are these results compare to? I suggest clarification here so that readers can understand “mush smaller” more clearly.
Some comment on minor points (typos, figure suggestions etc.)
- In figure 5, I suggest using lines instead of connected marks to present resistivity results, so that data presentation is clearer.
- In line 202, “what” could be “which”.
- Figure 7 panels could be arranged in more compact manner.
- In figure 8, a grey number in the bottom should be deleted.
- In figure 12, I suggest different composition to be labelled in legend instead of in figure caption.
- In line 257, 3.4 should be 3.6
- Panels in each figure need to be labelled.
- I suggest include this more recent neutron scattering article: DOI: 10.1088/0953-8984/28/14/146001
Author Response
We thank the Reviewers’ for their careful reading of the paper and for their constructive remarks. In order to take into account the latter, the paper has been revised. All changes are marked in yellow.
Reviewer #3:
- English language and style are fine/minor spell check required
A: The spelling has been checked.
- In the introduction part (line 35), the application of ZnCr2Te4 as a magnetoelectric compound merits more reference and introduction before discussing the underlying physics, so the article can be more accessible to non-specialist readers.
A: The introduction has been extended to include the following text: “Among seleno-spinel crystals such as AB2Se4 matrices, where A and B are the metallic ions occupying tetra- and octahedral sites, respectively, and matrices diluted with non-magnetic ions, due to the large cubic unit cell (about 10 Å), the effects of site disorder, lattice frustration and random distribution of spin interactions [1–5] create new potential applications.”; …, “An increasing dc magnetic field shifts TN to lower temperatures while a susceptibility peak in the paramagnetic region—to higher ones. Next, the first critical field Hc1 values connected with a metamagnetic transition decrease slightly with temperature while the values of the second critical field Hc2, connected with the breakdown of the conical spin structure, drop rapidly with temperature suggesting a spin frustration of the re-entrant type [2[. Low-angle neutron scattering (SANS) measurements showed the absence of any long-range magnetic order in the high-field (spin-nematic) phase, as well as the fact that all observed phase transitions were surprisingly isotropic concerning the field direction [11].”
- In table 1, authors used SEM/EDX to establish the chemical composition of the samples. For sample #2 to #6, the Zn composition are within 1% of the nominal doping and Ta composition are established accordingly, however for sample #1 the Zn is 5% more than nominal doping, which indicates the error bar of SEM is up to 5%, and it will render the Ta doping level up to 6% (0.12) in question. I do understand the XRD data shows systematic change of lattice parameters, I recommend authors either clarify the limit of SEM in a more detailed manner or even better, present ICP results.
A: We do not use the ICP method due to destroying of the sample during measurements. The determined real chemical composition showed a lower amount of incorporated tantalum than the nominal composition. It may be due to the lower amount of tantalum transported to the crystallization zone. On the other hand, the separation of tantalum from the system's during the dissolution and crystallization processes is likely.
- In part 3.4, resistivity and SQUID results are presented, however the setup of the measurement, especially the sample orientation information is missing. In parent compound, previous neutron scattering results indicate an incommensurate helical structure with different domains, under small magnetic field, domain selection occurs, so I naively figure this sample is not isotropic. Since the article is about single crystals, orientation of the measurement needs clarification.
A: In subsection 3.4, the following phrases were added: “The influence of anisotropy on the value of electrical resistivity in the studied spinel single crystals was not observed, as in the CuCr1.6V0.4Se4 spinel single crystal in which the resistance was measured in the direction of [001] and [111] [28].”, …, “No significant influence of the crystallographic direction on the value of the specific electrical resistance was observed. It may be because the spinel structure has a high cubic symmetry. The SANS measurements also confirmed the isotropic phase transitions concerning the field direction in the monocrystalline ZnCr2Se4 spinel [11].”
- In figure 8 and table 5, how are the HC1 and HC2 established? By taking first derivative or fitting etc.?
A: Chapter 2. Materials and Methods states that the derivative method dχ/dT determined the Néel temperature, TN, and the critical fields of HCl and HC2 against T and dM/dH against H., respectively.
- In part 3.5 where heat capacity measurement is presented, one main conclusion is that a phase transition shifting to lower temperature with increase field. Is this phase transition associated with the upper critical field? How does the result compare with undoped parent compound?
A: Generally, measurement of heat capacity and ac magnetic susceptibility in the presence of an increasing dc magnetic field causes a shift of the specific heat peak and the Néel temperature towards lower temperatures. This phenomenon is accompanied by a phase transition from cubic to the tetragonal structure. In other words, a strong dc magnetic field broadens the temperature range of the paramagnetic state and the cubic structure of the spinel under study. The upper critical field only informs that the sample has reached the state of ferromagnetic order. The situation is similar in the case of the ZnCr2Se4 matrix. Relevant text has been added in the conclusions.
- In line 246, could authors elaborate more on “much smaller than expected” a bit more? What is the underlying physics/reference are these results compare to? I suggest clarification here so that readers can understand “mush smaller” more clearly.
A: We are grateful Referee for the interesting comment due to the small total value of the measured entropy for the series of investigated spines, as well as suggestion that our data are obtained for the fields lower than Hc1. We explained the meaning of “mush smaller” entropy. The sentence is now more clear for readers: “For all investigated samples, the value of the magnetic and phonon contribution to the entropy at the ordering temperature TN and B = 0 is about 50% of the entropy expected, considering the magnetic contribution Sm = R×ln(2S+1) = 11.52 J/K molCr. The field dependence of C shown in Figure 12 is associated with the metamagnetic transition at the first critical field Hc1.”
Some comment on minor points (typos, figure suggestions etc.):
- In figure 5, I suggest using lines instead of connected marks to present resistivity results, so that data presentation is clearer.
A: Figure 5 has been corrected. The markers were replaced with lines.
- In line 202, “what” could be “which”.
A: The word "what" has been replaced by the word "which".
- Figure 7 panels could be arranged more compactly.
A: The panels in Fig. 7 are arranged more compactly. Similarly, the panels in Fig. 6.
- In figure 8, a grey number in the bottom should be deleted.
A: The grey number was not in the original drawing of Fig. 8.. The original drawing has been reinserted into the manuscript.
- In figure 12, I suggest different composition to be labelled in legend instead of in figure caption.
A: Figure 12 has been corrected.
- In line 257, 3.4 should be 3.6.
A: In line 257 the correct subsection number was inserted, i.e. 3.6.
- Panels in each figure need to be labelled.
A: The panels in the figures are labelled with lower case letters of the alphabet.
- I suggest include this more recent neutron scattering article: DOI: 10.1088/0953-8984/28/14/146001
A: In Conclusions4, the following text was added: “Generally, the shift of the specific heat peak and the Neel temperature [2-5,30] towards lower temperatures with increasing dc magnetic field is usually accompanied by a phase transition from cubic to the tetragonal structure. In other words, a strong dc magnetic field broadens the temperature range of the paramagnetic state and the cubic structure of the spinel under study. The upper critical field informs that the sample has reached the state of FM order below the structural transition. The situation is similar in the case of the ZnCr2Se4 matrix.”
Round 2
Reviewer 2 Report
It is unacceptable to pretend that corrections were made when they were not. The following comments were discarded:
- Abstract must be enriched via valuable results which pave the way for understanding the audiences. Also, please include your recommendations and future prospects. Please define the used abbreviations, while keeping them (example: X-ray diffraction (XRD), etc.).
- Formatting and quality of the figures should be considerably improved (the plots are not of the same size, some of them are too large.
- References. There is a tendency of SELF-citation. This should be reconsidered.
Also, please remove the references from the Conclusions sections.
If the manuscript will not be accordingly revised, I will not recommend its publication.
Author Response
The paper has been again revised. All changes are marked in yellow.
Reviewer #2:
- It is unacceptable to pretend that corrections were made when they were not.
A: The authors made a reasonable effort to improve the work, taking into account the comments of all Reviewers.
- Abstract must be enriched via valuable results which pave the way for understanding the audiences. Also, please include your recommendations and future prospects. Please define the used abbreviations, while keeping them (example: X-ray diffraction (XRD), etc.)
A: The abstract has been redrafted. Abbreviations were not used.
- Formatting and quality of the figures should be considerably improved (the plots are not of the same size, some of them are too large.
A: All the figures have been formatted, some of them additionally compacted and saved in the word.docx format, which allows them to be enlarged or reduced by the Technical Editorial Department.
- References. There is a tendency of SELF-citation. This should be reconsidered.
A: Self-citation is due to the fact that there are no publications from outside authors on critical magnetic fields. Most of the works discuss only the physico-chemical properties of the ZnCr2Se4 matrix and its solutions in poly-, mono- or nanocrystalline form.
- Also, please remove the references from the Conclusions sections.
A: The references from the Conclusions section have been removed.
- If the manuscript will not be accordingly revised, I will not recommend its publication.
A: We hope that this time, after these corrections and clarifications, the Reviewer's opinion will be positive.
Reviewer 3 Report
The authors have answers all my comments properly and I suggest the article to be published after proofreading.
A few subscripts that needs to be corrected (line 46,48,196,197,198,203,219). I also notice figures 6 and 7 are not that clear when I try to magnify them, maybe use eps/pdf source file.
Author Response
Reviewer #3:
- The authors have answers all my comments properly and I suggest the article to be published after proofreading.
A: The authors would like to thank the Reviewer #3 for a substantive and positive opinion.
- A few subscripts that needs to be corrected (line 46,48,196,197,198,203,219). I also notice figures 6 and 7 are not that clear when I try to magnify them, maybe use eps/pdf source file.
A: Subscripts have been corrected. The compact figures 6 and 7 can be enlarged since they are in word.docx format. The Technical Editorial Department should have no problem with that.